

# The feature enhancement method of artistic images based on histogram equalization and bilateral filtering

Wenjing Zhang

Art School, Zhengzhou University of Science and Technology, Zhengzhou, China

## ABSTRACT

To improve the rendering effect of artistic images, a method enhancing features of artistic images is proposed based on histogram equalization and bilateral filtering in the article. Firstly, artistic images are divided into both high and low-frequency representations, and the multi-step enhancement processing level is delimited by multi-band decomposition. Secondly, the noise in the image is removed by bilateral filtering. Then, the grey-level histogram of the image is modified by using the histogram equalization. Finally, the features of the artistic image are enhanced by global tone mapping after histogram equalization processing is conducted. Then, the image is sharpened to improve the enhancement effect further. The experiments show that the features of the color and edge details turn out to be more vivid and clearer after the proposed method is implemented. The structural similarity (SSIM) measure of the image increases to 0.973, and the average gradient gets close to 0.8, which shows that the proposed method is effective.

## INTRODUCTION

An artistic image is a graphical kind of symbol produced by artistic means, which is formed by the internal logic system of complex visual graphics. Its elements are mainly composed of modeling points, lines, surfaces, bodies, blocks, colors, textures, *etc.*. It is relatively different from the original graphics in nature (*Zheng, 2023*). This image is a visual form created by certain requirements and modeling rules. Distinct from practical images, artistic images follow more subjective and aesthetic expression with more artistic and creative forms (*Guo, 2023*).

The main reason for implementing feature enhancement on artistic images is to highlight the effective interesting features to be processed while diluting or removing unwanted information. To enhance the features of the image, its visual effect and clarity can be improved, and the visual perception and aesthetic needs of human beings can be better met (*Chen & Yang, 2022*; *Kobayashi et al., 2022*). At the same time, feature enhancement can also provide more representative information for subsequent image analysis and processing and thus improve the accuracy and efficiency of image processing.

Corresponding author
Wenjing Zhang,
meilandusci@163.com

For example, a method enhancing image features is proposed based on a convolutional network (CNN) model, which uses an image color model and denoising autoencoder to complete the pre-processing of low-illumination images (*Zhou & Li, 2023*). Then, the image mapping function is designed by the piecewise transformation method, and the objective function of the image enhancement is obtained. Finally, the CNN model and the corresponding loss function are constructed to complete the image feature enhancement process. An image enhancement method is proposed by combining feature fusion and physical correction (*Wand et al., 2023*). Self-building blocks are used to replace the feature fusion network of the encoder and decoder structure of the convolutional layer to correct the color bias of the image. The improved feature fusion module in the network reduces the damage of the full connection layer to the image spatial structure, thus protecting the spatial features, and reducing the number of parameters of the module. Then, the improved attention module is used to extract texture details and protect background information through parallel pooling computations. Finally, the multi-color model correction module is employed to correct the relationship between pixels to further reduce color biases and improve contrast and brightness features. An image enhancement method is proposed based on multi-level feature fusion that utilizes multiscale sampling to construct a U-shaped network and introduces multiple attention mechanisms to multi-thread image flow. The feature vectors of each branch interact across channels to cooperatively and gradually suppress redundant information. Then, the feature fusion module is employed to enhance the perception of low-scale texture details and multi-level features, respectively. Finally, a loss function composed of peak signal-to-noise ratio and structural similarity (SSIM) index is designed to guide the network to learn the mapping relationship between images from shallow to deep structures, thus speeding up the convergence of the model and helping improve the model performance and image enhancement, respectively.

However, it is found that SSIM and average gradient of artistic images become low after processing when the conventional algorithms and the enhancement results are not ideal in practical applications. To resolve this problem, a new feature enhancement method for artistic images is designed based on histogram equalization and bilateral filtering. The proposed method is expressed as follows:

Firstly, the form of multi-stage processing is adopted to delimit the multi-stage enhancement processing level of the image by multi-band decomposition based on dividing the high and low-frequency representations of an artistic image. To fuse different scales of the image, the structure information and texture features of the image can be described better, which provides more powerful support for the subsequent feature enhancement processing. Secondly, the artistic image is denoised by bilateral filtering. Then, the histogram equalization method is employed to correct the gray level histogram of the image, so that the pixel intensity distribution of the image becomes more uniform and the contrast of the image is improved in the end. Finally, the features of the artistic image are enhanced by global tone mapping. On this basis, the image is sharpened to highlight the feature edges and details to further improve the enhancement effect after histogram equalization processing is run.

The rest of the article is outlined as follows: 'The design of a method used for enhancing the features of the artistic image' presents the proposed method. 'Experiment and the Analysis of the Results' is allocated to the experiment and the analysis of the results. 'Conclusion' presents the conclusion.

# THE DESIGN OF A METHOD USED FOR ENHANCING THE FEATURES OF THE ARTISTIC IMAGE

## High and low-frequency divisions of an artistic image and the level demarcation of the multi-level enhancement processing

Based on dividing the high and low-frequency representations of artistic images, this study adopts the form of multi-stage processing and delimits the multi-stage enhancement processing level by multi-band decomposition.

Usually, it is necessary to divide the high-frequency and low-frequency parts of the image before processing an artistic image. The high-frequency part mainly contains the details of the image, such as edge, texture, *etc.* The low-frequency part generally includes the structural information of the image, such as contour, shape, *etc.* Each frequency band corresponds to an enhanced processing level. Each band or level can be enhanced to highlight the details of that band or level (*Li et al., 2023*; *Zhang et al., 2024*).

Based on the processing requirements of the actual artistic image, the current transfer function is measured and calculated, and the best control is generally found between 0.35 and 0.66. Then, according to the change in the transfer function of the ideal low-pass filter, the cutoff frequency is measured and calculated in Eq. (1)

$$r = t^2 - \sum_{B=1} \varpi B + \gamma \qquad (1)$$

where $r$ represents the cut-off frequency; $t$ stands for lossless frequency; $\varpi$ represents directional high and low frequencies; $B$ indicates the calibration of the covered area; $\gamma$ means the converted mean. When combined with the current measurements, the cutoff frequency is calculated (*Seungchul, Hyunjin & Tara, 2023*). On this basis, the image is divided according to the brightness, contrast, color, and other features, and the multi-level enhancement processing level is designed. The content of the specific division is shown in Fig. 1.

When combined with Fig. 1, the division of multi-level enhancement processing levels is realized. The hierarchical delineation of multi-level enhanced processing is to construct multiscale image representation. To process an image, multiscale analysis is an important method, which can extract feature information at different scales by processing images at different scales (*Kwon et al., 2023*; *Ding et al., 2023*). The image can be divided into several different scales and each scale can be enhanced to obtain a multiscale image representation.

The structure information and texture features of the image can be better described by fusing the different scales of the image, which provides more powerful support for the subsequent feature enhancement processing.

| Brightness of images | Contrast of images | Color of images |
|---|---|---|
| Image brightness area | Image purity | Brightness of the image |
| Image junction position | Image clarity | Saturation of an image |
| Image area proportion | Image comparison position | Color distribution of images |

**Figure 1** **Artistic image multilevel enhancement processing hierarchy classification diagram.**

## Bilateral filtering processing of the artistic image

In the above process, the visual quality of artistic images can be improved and the artistic effect can be enhanced through the division of high and low frequencies and the demarcation of multi-level enhancement processing levels. On this basis, the denoising of artistic images is realized by employing bilateral filter processing.

Bilateral filtering is a kind of filter that can retain edge information and denoise at the same time. The reason is that the denoising effect can be achieved with the filter composed of two functions and the filter coefficient is determined by the geometric space distance and the pixel difference, respectively (*Zeng, 2023*; *Liu, S. & Tao, 2023*). The two-sided filter has one more Gaussian variance than the Gaussian filter, which is a Gaussian filter function based on spatial distribution, so the pixels far away from the edge will not have a greater impact on the edge pixels to avoid the edge blur (*Yang & Zhou, 2023*).

The bilateral filtering is defined in Eq. (2):

$$F_I = \frac{\sum_{i,j=-w}^{w}(G_{\sigma 1}(x,y) x G_{\sigma 2}(x,y) x I(x,y))}{\sum_{i,j=-w}^{w} G_{\sigma 1}(x,y) x G_{\sigma 2}(x,y)} \tag{2}$$

where $I$ represents the artistic image to be processed; $F_I$ represents the filtered image, and $G_{\sigma 1}(x,y)$ represents the Gaussian kernel function, representing the spatial similarity of points with $(x,y)$ as the center and $w$ as the radius; $\sigma_1$ represents the variance parameter; $G_{\sigma 2}(x,y)$ represents the pixel similarity of points with $(x,y)$ as the center and $w$ as the radius; $\sigma_2$ represents the variance parameter.

The gray value $h_I$ of the artistic image after bilateral filtering is expressed in Eq. (3):

$$h_I = \frac{\sum_{(x,y)\in n} w_d(x,y) w_h(x,y) gray(x,y)}{\sum w_d(x,y) w_h(x,y)} \tag{3}$$

where $w_d(x,y)$ represents the spatial weight of the image; $w_h(x,y)$ represents the gray similarity weight of the image; $gray(x,y)$ represents the grayscale value of the image at point $(x,y)$.

The expression of the space weight $w_d(x,y)$ is presented in Eq. (4):

$$w_d(x,y) = exp(-\frac{|o-x|^2 + |k-y|^2}{2\varsigma_d^2}) \tag{4}$$

where $\varsigma_d$ represents the standard deviation of the space domain; $(o,k)$ represents any point in the image other than $(x,y)$.

The similarity weight of the image grayscale $w_h(x,y)$ is expressed in Eq. (5):

$$w_h(x,y) = exp(-\frac{|gray(o,k) - gray(x,y)|^2}{2\varsigma_d^2}) \tag{5}$$

where $\zeta_d$ represents the standard deviation of grayscale; $gray(o,k)$ denotes the grayscale value of the image at point $(o,k)$.

The analysis suggests that the bilateral filtering process is affected by the spatial weight and grayscale similarity weight, and there is a linear relationship between the standard deviation of the spatial domain and the radius of the filtering window, which will affect the sharpness of the artistic image after the bilateral filtering is run (*Wu et al., 2023*). Since more than 95% of the components of the Gaussian function are concentrated in the interval $[-2\zeta_d, 2\zeta_d]$, the standard deviation $\zeta_d$ of the space domain meets Eq. (6) to ensure the clarity of the image:

$$\zeta_d = \frac{al}{2} \tag{6}$$

where $I$ represents the radius of the filtering window; $a$ represents the constant, and finally, the optimal range of constant changes between 0.80–0.95 through many tests, which can effectively prevent image blurring.

In addition, the gray standard deviation has a greater impact on the effect of the bilateral filtering. Increasing the gray standard deviation can improve the denoising effect, but at the same time, the edge details of the image may be lost (*Gong et al., 2022*). A linear relationship exists between gray standard deviation and noise variance in the ratio ranging from 2–3. After running many tests, $\zeta_h = 2\zeta_x$ is taken. The noise variance $\zeta_x$ obtained by the Laplace transform is expressed in Eq. (7):

$$\zeta_x = \sqrt{\frac{\pi}{2}} x \frac{\sum |gray(o,k) xM|}{(X-1)(Y-1)} \tag{7}$$

where $X$ represents the image width, $Y$ stands for image height, and $M$ denotes the discrete Laplace transform mask.

Thus, the spatial standard deviation and gray standard deviation of the optimized bilateral filter are obtained, and the denoising process of the artistic image is completed.

## Histogram equalization process of the artistic image

After image denoising, histogram equalization is used to process the artistic image globally, which can enhance the overall contrast of the image and make it more vivid. Histogram equalization can make the distribution of pixel intensity more uniform and improve the image contrast by correcting the gray-level histogram (*Liu et al., 2022*; *Wang, Zhang &*

*Zhang, 2022*). This method is especially suitable for those artistic images with low contrast, which can significantly enhance their detail and color levels.

Generally, the gray value of each pixel in the image is specified to be between [0,255], and the histogram of the image is represented by a discrete function $f(h_k) = n_k$, where $h_k$ represents the gray value $n_k$ of any pixel in the image and the number of pixels with gray value $h_k$ in the image.

The gray value of the image can be delineated by the discrete function, and the histogram shows the distribution of the image's gray value. In the equalization, the histogram is generally normalized and then the gray value is reassigned.

Assuming that the infrared image is represented by the tensor *MxN* and the number of image pixels is counted *MN*, the normalized histogram can be expressed by Eq. (8):

$$p(h_k) = \frac{n_k}{MN} \tag{8}$$

where it represents the estimation of the probability that the gray value of the image is $h_k$. In other words, the normalization method turns all components of the histogram equal to 1 after the addition.

Based on the above analysis, the processing steps for setting up artistic image histogram equalization are as follows:

Step 1: The image is divided into two regions. The first region is the low-illuminance region with a threshold $L_L$, and the second one is the high-illuminance region with a threshold $L_H$. Then, the image histogram is divided and the sub-histograms of the three images are obtained.

Step 2: Establish the probability density function corresponding to the sub-histogram, and modify the value of the function;

Step 3: According to the correction results, the corrected cumulative distribution function is established;

Step 4: The image equalization process is completed by combining the modified cumulative distribution function with the corrected output function curve of gamma correction.

The local features and illumination conditions of artistic images are dynamically adjusted, and the threshold of low illumination and high illumination areas can be determined according to the key values. The expression for the key value is given in Eq. (9):

$$key = \frac{\bar{L} - L_{min}}{Wx(L_{max} - L_{min})} \tag{9}$$

where $L_{min}$ and $L_{max}$ represents the lower and upper limits of image brightness, respectively; $\bar{L}$ represents the average brightness of the image; $w$ represents the energy of an image pixel.

Based on the key value, the thresholds $L_L$ and $L_H$ are determined in Eq. (10):

$$\begin{cases} L_L = L_{max} - \dfrac{\bar{L}(L_{max} - L_{min})}{key} \\ L_H = L_{max} - \varepsilon xkey(L_{max} - L_{min}) \end{cases} \tag{10}$$

where $\varepsilon$ represents the brightness compensation value.

Then, to improve the visibility of the image, the image histogram is divided according to the threshold results, and three sub-histograms $I_1$, $I_2$ and $I_3$ are obtained, respectively.

To avoid edge effects in histogram equalization processing, it is necessary to adjust the probability density of sub-histograms to achieve the final probability density function given in Eq. (11):

$$\rho = \frac{g_{max} \cdot g_{min}}{(L_H - L_L) . \beta} \tag{11}$$

where $g_{max}$ and $g_{min}$ represents the upper and lower limits of the function values; $\beta$ indicates the adjustment parameters.

Since there may be errors in the obtained function value, it needs to be corrected. The corrected high-precision probability density function is given in Eq. (12):

$$\rho' = \frac{\sum_{z=1}^{max} \rho}{n_j} \tag{12}$$

where $n_j$ represents the number of pixels in the sub-histogram. Since the area with the smaller gray level in an artistic image is usually the foreground pixel, the one with the larger gray level is usually the background pixel. Therefore, according to the modified probability density function, the gamma correction method is used to stretch the pixel level of the sub-histogram of the image, to complete the histogram equalization processing of the artistic image.

## Enhancing artistic image features based on global tone mapping

Based on the artistic image after histogram equalization processing, the features of the artistic image are enhanced by employing global tone mapping, which can be regarded as a process of gray stretching given in Eq. (13):

$$R = \frac{A(x,y)}{V(x,y) + G(x,y) + A(x,y)} \tag{13}$$

where $A(x,y)$ represents the ratio of the pixel value of the image to its mean value; $V(x,y)$ represents the local contrast operator, which can stretch the contrast of the image in the local neighborhood to highlight the details of the image; $G(x,y)$ represents the global contrast operator, which can enhance the gray level of the darker area and enhance the contrast of the image.

Equation (13) comprehensively considers the global and local contrasts of the artistic image, and the distribution range of pixel values in the high-brightness region is reduced in the stretching process, while the distribution range of pixel values in the low-brightness region is expanded, thus achieving the effect of enhancing image details (*Xu et al., 2022*).

Equation (14) delineates how $A(x,y)$, $V(x,y)$ and $G(x,y)$ are computed:

$$\begin{cases} A(x,y) = \dfrac{h_I}{h} \\ V(x,y) = exp(-\delta \dfrac{A(x,y)}{\bar{g}}) \\ G(x,y) = exp(-\lambda \dfrac{\bar{h}}{\bar{g}}) \end{cases} \tag{14}$$

where $\bar{h}$ represents the average value of image pixels; $\delta$ stands for local contrast stretch factor; $\bar{g}$ represents the Gaussian mean of pixel scores in the local neighborhood of the target pixel point; $\lambda$ represents the enhancement coefficient of the global contrast operator.

The global contrast of the image decreases with the increase in $\lambda$ score. When the $\lambda$ increases, more details can be seen in the area with a lower gray level of $\lambda$, whereas when the A-value decreases, many details of the image cannot be displayed.

## Full-color sharpening of artistic images

In this study, after the feature enhancement processing of artistic images, the image is further sharpened to full-color processing. Sharpening can help highlight the feature edges and details of the image, further improving the enhancement effect.

Panchromatic sharpening ensures that all color channels of the image are evenly sharpened, avoiding color distortion or loss of details. Panchromatic sharpening before feature enhancement ensures the quality and consistency of the input image, thus better serving subsequent image analysis and processing tasks, respectively. The local correlation coefficient (LCC) is used to measure the similarity of spectral features between artistic images and panchromatic images in the low-frequency part, and the fourth-order correlation coefficient (FOCC) between the two images is calculated, and the fusion coefficient is determined by comparison among them.

After the fusion processing of panchromatic image and artistic image, if the LCC score of the low-frequency part gets low, it means that the spectral feature similarity score of each pixel becomes also low and cannot be replaced. On the contrary, if the local correlation coefficient between the artistic images gets large, it indicates that there are a large number of similar spectral features in the two artistic images, which can effectively avoid the appearance of image distortion. Equation (15) is used to calculate the local correlation coefficient:

$$LCC_s = \frac{K_{a,b}(x,y)}{\sqrt{K_a(x,y)K_b(x,y)}} \tag{15}$$

where $K_{a,b}(x,y)$ represents the local covariance between any two image blocks $a$ and $b$ centered on pixel point $(x,y)$; $K_a(i,j)$ and $K_b(i,j)$ represent the local variances of image blocks $a$ and $b$.

To obtain a more satisfactory panchromatic sharpening effect, the scores of LCCs and FOCC are compared to determine whether the high-frequency coefficient needs to be replaced. The four-order correlation coefficient evolved based on the correlation coefficient among them, which can more accurately measure and describe the SSIM of artistic images. The fourth-order correlation coefficient is given by Eq. (16):

$$FOCC_{a,b} = \frac{1}{mxn} x \frac{\sum_{x=1}^{X}\sum_{y=1}^{Y}(B(x,y)-\tau^a)^2(E(x,y)-\tau^b)^2}{\sqrt{\sum_{x=1}^{X}\sum_{y=1}^{Y}(B(x,y)-\tau^a)^2(E(x,y)-\tau^b)^2}} \tag{16}$$

where $B(x,y)$ and $E(x,y)$ represent the image matrix of size $X \times Y$, $\tau_a$ and $\tau_b$ represent the mean of the matrix.

The steps of the panchromatic sharpening of the artistic image are given as follows:

Step 1: The artistic image is processed by the Curvelet interpolation method, and NSCT transformation is developed to obtain the high and low-frequency coefficient matrix of the artistic image, and then a new high and low-frequency coefficient matrix is obtained by NSCT transformation.

Step 2: For the low-frequency coefficient matrix after the decomposition of the artistic image and panchromatic image, $(x, y)$ is selected as the center, the radius is set to 2, and the local correlation coefficient and fourth-order correlation coefficient of the image block are calculated by Eqs. (15) and (16), respectively.

If the local feature correlation between the target region and the artistic image gets relatively high, the high-frequency coefficient of the panchromatic image at the point is employed to replace the high-frequency coefficient of the corresponding point $(x, y)$ of the artistic image, while effectively ensuring that there will be no spectral distortion. On the contrary, the high-frequency coefficient of the artistic image is kept unchanged.

Step 3: Expand the NSCT transformation of the low-frequency coefficient and the high-frequency coefficient of the artistic image after processing, and obtain the artistic image after the fusion processing is run.

Step 4: Expand all the fused images in the RGB color space, and then realize the artistic image full-color sharpening processing and realize the artistic image feature enhancement processing.

# EXPERIMENT AND THE ANALYSIS OF THE RESULTS

To verify the practical application performance of the proposed method used for enhancing artistic image features based on histogram equalization processing and bilateral filtering, the following experiments are designed.

## Experimental design

OpenCV was used as the simulation environment. OpenCV is a widely used open-source computer vision library that provides rich image processing and computer vision functions and supports histogram equalization, edge detection, and filter processing of different specifications, which is suitable for this experiment.

The experimental data was gathered from an online platform focused on contemporary art called the Art Encyclopedia database, which provides a large number of images and related content of contemporary artworks. In this database, users can search and browse a variety of contemporary artworks. In this study, 200 artistic images were randomly selected from the Art Encyclopedia database for the experiment, and the image size was adjusted to $2,500 \times 3,000$ with 2 bytes of pixels.

To avoid the uniformity of experimental results, the methods in *Zhou & Li (2023)* and *Wand et al. (2023)* are employed to compare performance with the proposed method.

## Results

Firstly, the two artistic images in Fig. 2 are taken as examples to visually compare the effects of the feature enhancement processing in different methods.

The processing results of feature enhancement in Fig. 2 by different methods are shown in Fig. 3.

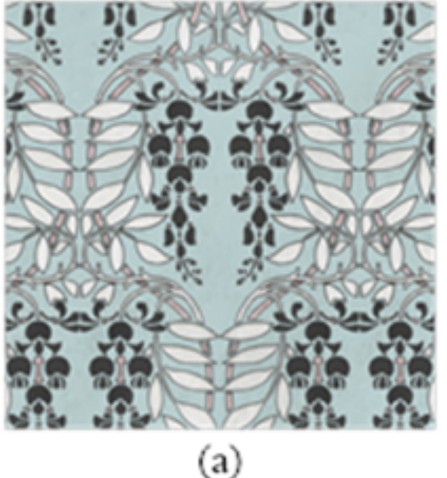
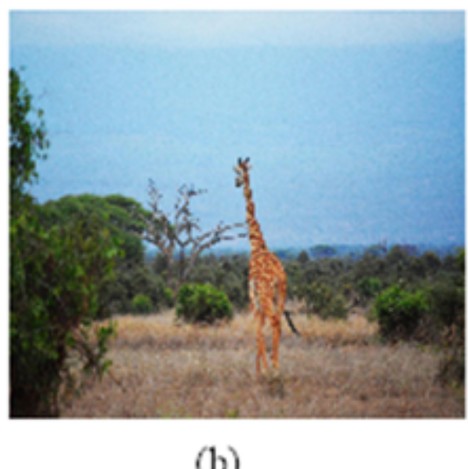

(a)                                                    (b)

**Figure 2** **(A–B) Sample image.** Image source credits: Rawpixel, https://www.rawpixel.com/image/2421383/free-illustration-png-art-deco-seamless-pattern. Giraffe, https://www.pexels.com/photo/photograph-of-giraffe-1319515.

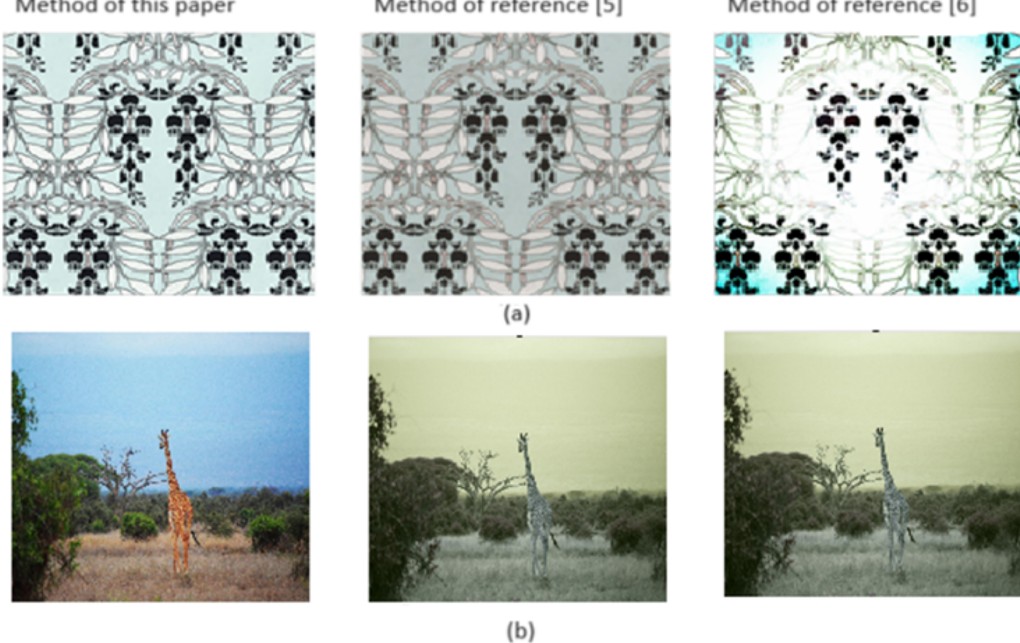

**Figure 3** **(A–B) Feature enhancement processing results.** Image source credits: Rawpixel, https://www.rawpixel.com/image/2421383/free-illustration-png-art-deco-seamless-pattern. Giraffe, https://www.pexels.com/photo/photograph-of-giraffe-1319515.

To compare Figs. 2 and 3, the color features and edge detail features of the image get more vivid and clear after the proposed method is implemented. However, when the methods in *Zhou & Li (2023)* and *Wand et al. (2023)* are applied, the images have different degrees of noise and fading, which cannot effectively reflect the characteristics of artistic images.

To further verify the image feature enhancement performance of different methods, SSIM, and average gradient are employed as indicators.

For example, SSIM can be implemented to quantify the retention degree of feature details of artistic images after different methods are applied. Equation (17) presents how the SSIM is calculated:

$$SSIM = (\frac{2 \times nQ_1 \times nQ_2 + C_1}{Q_1^2 + Q_2^2 + C_1} \times \frac{2 \times \zeta Q_1 \times \zeta Q_2 + C_2}{Q_1^2 + Q_2^2 + C_2} \times \frac{\zeta Q_1 Q_2 + C_3}{Q_1 Q_2 + C_3})^{\mu} \qquad (17)$$

where $Q_1$ and $Q_2$ represent the original image and the enhanced image respectively; $nQ_1$ and $nQ_2$ represent the means of pixels of $Q_1$ and $Q_2$ respectively; $\zeta Q_1$ and $\zeta Q_2$ represent the pixel standard deviation of $Q_1$ and $Q_2$, respectively; $\zeta Q_1 Q_2$ represents the pixel covariance of $Q_1$ and $Q_2$; $C_1$, $C_2$, and $C_3$ represent constants, which are implemented to avoid cases where the denominator is zero, and $\mu$ is a constant in the range $[0,1]$, which controls the importance of brightness to structural similarity.

When the SSIM is calculated, the image is generally divided into a series of non-overlapping small blocks, and the SSIM score of each small block is calculated. Then, the SSIM scores of all small blocks are averaged to get the final SSIM score. The SSIM changes in $[0,1]$. The closer the value is to 1, the more similar the structure of the two images is and the better the quality is.

After applying different methods, the SSIM scores of artistic images are shown in Table 1.

Table 1 depicts that as the time increases in the experiment, the SSIM of the artistic images after feature enhancement processing also changes with the applications of different methods. After the method in *Zhou & Li (2023)* is applied, the SSIM of the images changes between 0.862 to 0.896 and reaches the maximum score in 50 tests. After the method in *Wand et al. (2023)* is applied, the SSIM of images ranges between 0.833–0.885 and reaches the maximum score at 40 tests. After the proposed method is implemented, the SSIM of the images ranges between 0.936 to 0.973 and also reaches the maximum score at 40 tests. Based on the above tests, after the proposed method is applied, the SSIM of the image gets higher, indicating the enhancement processing effect of the proposed method.

The average gradient can reflect the clarity of the image and the change of details after feature enhancement is run. In general, the larger the average gradient, the more prominent the image texture features and the better the detail retention effect. Eq. (18) presents how it is computed:

$$\xi = \frac{\sum_{x=1}^{X}\sum_{y=1}^{Y}\sqrt{(f(x-1,y)-f(x,y))^2 + (f(x,y-1)-f(x,y))^2}}{(X+1)(Y+1)} \qquad (18)$$

**Table 1** Statistical result of SSIM value of artistic image.

| Number of tests | SSIM | | |
|---|---|---|---|
| | Method of this paper | Method of reference (*Zhou & Li, 2023*) | Method of reference (*Yang & Zhou, 2023*) |
| 10 | 0.936 | 0.864 | 0.835 |
| 20 | 0.940 | 0.891 | 0.858 |
| 30 | 0.951 | 0.874 | 0.869 |
| 40 | 0.973 | 0.862 | 0.885 |
| 50 | 0.958 | 0.896 | 0.833 |

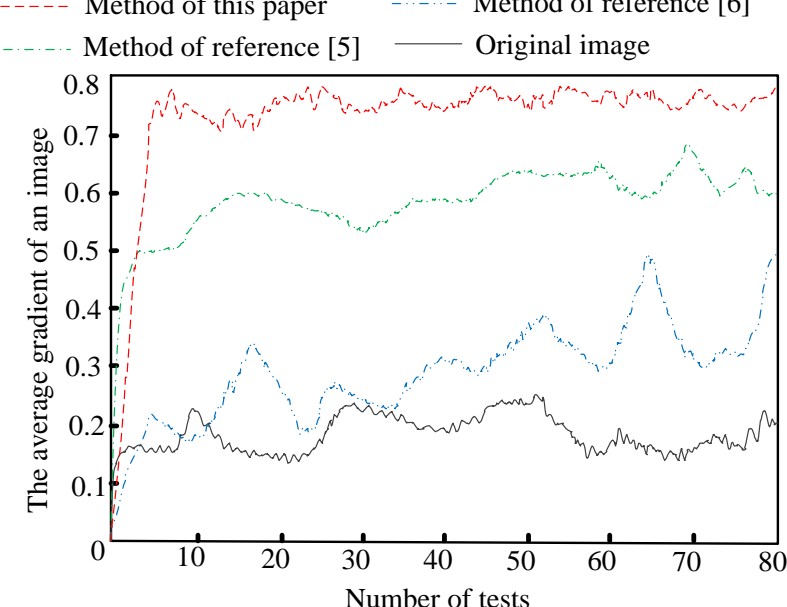

**Figure 4** Statistical results of image mean gradient.

where $f(x, y)$ represents the pixel value corresponding to the row $x$ and column $y$ of the image; $f(x-1, y) - f(x, y)$ and $f(x, y-1) - f(x, y)$ represents the first-order difference of $f(x, y)$ under $x$ or $y$, respectively.

After applying different methods, the average gradient of artistic images is shown in Fig. 4.

Figure 4 depicts that after the proposed method is implemented, the average gradient curve of the artistic image after feature enhancement processing is always higher than the average gradient curve of the two comparison methods and the original image, and the maximum score is close to 0.8, indicating that after the proposed method is implemented, The clarity of details in artistic images becomes much higher.

## CONCLUSION

A new feature enhancement method for artistic images is designed based on histogram equalization and bilateral filtering. The proposed method is expressed as follows:

Firstly, the form of multi-stage processing is adopted to delimit the multi-stage enhancement processing level of the image by multi-band decomposition based on dividing the high and low-frequency representations of an artistic image. To fuse different scales of the image, the structure information and texture features of the image can be described better, which provides more powerful support for the subsequent feature enhancement processing. Secondly, the artistic image is denoised by bilateral filtering. Then, the histogram equalization method is employed to correct the gray level histogram of the image, so that the pixel intensity distribution of the image becomes more uniform and the contrast of the image is improved in the end. Finally, the features of the artistic image are enhanced by global tone mapping. The image is sharpened to highlight the feature edges and details to further improve the enhancement effect after histogram equalization processing is run.

Experimental results show that the proposed method is effective in improving the image quality. The image SSIM index becomes as high as 0.973, and the average gradient is close to 0.8.

The limitation of the research is based on a limited number of samples. More data-based investigations should be run and compared to get more reliable outcomes. Also, the variety of images should be increased.

Future research will focus on more data samples and varied samples to derive more insights and better images.

### Funding
The authors received no funding for this work.

### Competing Interests
The authors declare there are no competing interests.

### Author Contributions
- Wenjing Zhang conceived and designed the experiments, performed the experiments, analyzed the data, performed the computation work, prepared figures and/or tables, authored or reviewed drafts of the article, and approved the final draft.

### Data Availability
The data and code are available in the Supplemental Files.

### Supplemental Information
Supplemental information for this article can be found online at http://dx.doi.org/10.7717/peerj-cs.2109#supplemental-information.

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
