# Peer review of "The feature enhancement method of artistic images based on histogram equalization and bilateral filtering"

_PeerJ Computer Science, doi:10.7717/peerj-cs.2109_

## Round 0.1 · original submission · Major Revisions

Dear authors,

Thank you for submitting your article. Feedback from the reviewers is now available. It is not recommended that your article be published in its current format. However, we strongly recommend that you address the issues raised by the reviewers, especially those related to readability, experimental design and validity, and resubmit your paper after making the necessary changes.

Best wishes,

·

Basic reporting

Your work, An Artistic Image Feature Enhancement Method Based on Histogram Equalization and Bilateral Filtering, has caught my attention and I believe it makes a significant contribution to the field of image processing. The authors of this study provide a systematic way of feature enhancement in the multi-step enhancement procedure. This method involves dividing the image into high and low frequencies and then performing multi-band decomposition. With this technique, the overall boosting effect is improved by carefully adjusting different levels of image properties.

However, I would like to recommend a few changes to increase the caliber and clarity of your manuscript:

Although the rationale behind applying feature enhancement to artistic photographs is mentioned in the introduction, a more thorough explanation of the precise aims or purposes of the suggested approach would be beneficial. A clearer explanation of the intended results or enhancements anticipated by the suggested strategy would help readers comprehend its goal.

It would be easier to separate the suggested strategy from other methods if it focused on its original contributions or advances. It would be more appealing and significant if it was made clear what makes the suggested method unique and how it advances the area.

A succinct synopsis or road map of the following sections of the paper should be included at the end of the introduction to help readers get started and give them an idea of how the content will be organized and structured. This would make the paper easier for readers to navigate and help with comprehension.

Extend the criteria that were applied to the 200 creative photos that were chosen from the Art Encyclopedia database. Moreover, explain any image preprocessing (normalization, scaling, etc.) that was done prior to the experiments in order to guarantee validity and consistency in the analysis.
Provide a more thorough analysis of these outcomes after SSIM and average gradient values are shown in Table 1 and Figure 4. Describe the relationship between the improved images' visual quality and detail preservation and the observed changes in SSIM and average gradient values. This would make it easier for readers to comprehend the importance of the numerical data when evaluating the effectiveness of the suggested strategy.


Recognize any shortcomings or restrictions in the experimental design and methods, including any potential biases in the choice of images or limitations on the amount of computing power available. Outline possible directions for further study or development to overcome these drawbacks and raise the effectiveness and applicability of the suggested approach even further. This would show a critical understanding of the study's boundaries and add to the current conversation in the area.

Experimental design

All comments are provided in section 1.

Validity of the findings

No comments

Additional comments

No comments

Reviewer 2 ·

Basic reporting

The abstract succinctly outlines the purpose, methods, key findings, and implications of the study. It could be enhanced by briefly mentioning the comparative performance of the proposed method against existing methods to highlight its novelty and effectiveness right at the outset. The keywords are relevant but could be expanded to include "digital art enhancement" or "visual art algorithms" to capture a broader scope of potential scholarly searches.
Minor grammatical revisions may be needed to enhance readability. Consider having the manuscript professionally proofread or using software tools to ensure language precision.

Experimental design

The methodology is detailed and informative. However, including more explicit details about the software and settings used during the experiments (e.g., version numbers, specific parameter settings) would improve reproducibility. A small subsection on the limitations of the method would also prepare the reader for understanding the scope of application.

Validity of the findings

Incorporating a few qualitative comments from subjective assessments (if available) could enrich the discussion on image quality perception. Discussions about potential biases in the dataset or experimental setup should be included to provide a more balanced view of the results.

Additional comments

Incorporating visual comparisons of the image enhancements directly in the text, alongside more detailed statistical analysis, would provide readers with a clearer understanding of the method's efficacy. Additionally, expanding on the practical implications of the technique in real-world applications such as digital archiving or online art platforms in the introduction could strengthen the motivation for this research.

A dedicated discussion on potential biases and the limitations of the current methodology would provide a balanced view and suggest areas for further investigation. Enhancing the conclusion with direct links to future work and potential improvements could provide continuity for subsequent research efforts.

The conclusions are aptly drawn; however, adding a few sentences about potential future work directly linked to overcoming any limitations discussed earlier would provide a more rounded ending to the paper.

By addressing these minor revisions, the manuscript could offer a more robust and engaging contribution to the field of digital image processing for artistic applications.

Reviewer 3 ·

Basic reporting

An artistic image feature enhancement method is proposed based on histogram equalization and bilateral filtering. The author claim that color features and edge detail features of the image are more clear after the feature enhancement method. However, there are many studies in the literature focusing on histogram equalization and bilateral filtering e.g.
Jianbin Xiong, Dezheng Yu, Qi Wang, Lei Shu, Jian Cen, Qiong Liang, Huanyang Chen, Baocheng Sun, "Application of Histogram Equalization for Image Enhancement in Corrosion Areas", Shock and Vibration, vol. 2021, Article ID 8883571, 13 pages, 2021. https://doi.org/10.1155/2021/8883571

Li, Yu, Yuan, Z., Zheng, K., Jia, L., Guo, H., Pan, H., Guo, J., Huang, L.: A novel detail weighted histogram equalization method for brightness preserving image enhancement based on partial statistic and global mapping model. IET Image Process. 16, 3325–3341 (2022). https://doi.org/10.1049/ipr2.12567

On lines 28 - 29, the written sentence seems to be incomplete. There should be explicit statements defining the motivation of the research. There is a text repetition on lines 60 – 61 & Lines 74 – 76 & Lines 103 – 105. In Section 2.2, 2.3 and 2,4, it seems that we are learning but not reading a research paper. Methodology diagram should be there to reflect the methodology followed. Overall, the paper needs improvement.

Experimental design

The experimental design fit the aims and scope of the paper. However, the dataset is too small.

Validity of the findings

Lastly, the conclusion section lacks a detailed description of future research and limitations of the paper.

---

## Round 0.2 · accepted · Accept

Dear authors,

Thank you for the revision and for clearly addressing all the reviewers' comments. I confirm that the paper is improved. Your paper is now acceptable for publication in light of this revision.

Best wishes,

·

Basic reporting

All my previous revisions are addressed

Experimental design

OK

Validity of the findings

OK

Additional comments

All my previous suggestions are well addressed.

Reviewer 2 ·

Basic reporting

The structure ensures that all necessary details are conveyed succinctly and professionally, maintaining clarity and respect for the author's contributions.

Experimental design

This format ensures that you cover all critical aspects of your experimental design, providing enough detail for replication and understanding. It also helps in maintaining scientific rigor and transparency throughout your research documentation.

Validity of the findings

The validity of the findings is strengthened by rigorous adherence to methodological protocols and by the reproducibility of results across varied samples and settings.

Reviewer 3 ·

Basic reporting

Satisfactory .... Meeting all the requirements

Experimental design

Satisfactory

Validity of the findings

Satisfactory